# Urine-HILIC: Automated Sample Preparation for Bottom-Up Urinary Proteome Profiling in Clinical Proteomics

**DOI:** 10.3390/proteomes11040029

**Published:** 2023-09-28

**Authors:** Ireshyn Selvan Govender, Rethabile Mokoena, Stoyan Stoychev, Previn Naicker

**Affiliations:** 1NextGen Health, Council for Scientific and Industrial Research, Pretoria 0001, South Africa; 2ReSyn Biosciences, Edenvale 1610, South Africa; 3School of Molecular and Cellular Biology, University of the Witwatersrand, Johannesburg 2193, South Africa

**Keywords:** automated sample preparation, clinical proteomics, SWATH-MS (DIA), urinary proteomics, HILIC

## Abstract

Urine provides a diverse source of information related to a patient’s health status and is ideal for clinical proteomics due to its ease of collection. To date, most methods for the preparation of urine samples lack the throughput required to analyze large clinical cohorts. To this end, we developed a novel workflow, urine-HILIC (uHLC), based on an on-bead protein capture, clean-up, and digestion without the need for bottleneck processing steps such as protein precipitation or centrifugation. The workflow was applied to an acute kidney injury (AKI) pilot study. Urine from clinical samples and a pooled sample was subjected to automated sample preparation in a KingFisher™ Flex magnetic handling station using the novel approach based on MagReSyn^®^ HILIC microspheres. For benchmarking, the pooled sample was also prepared using a published protocol based on an on-membrane (OM) protein capture and digestion workflow. Peptides were analyzed by LCMS in data-independent acquisition (DIA) mode using a Dionex Ultimate 3000 UPLC coupled to a Sciex 5600 mass spectrometer. The data were searched in Spectronaut™ 17. Both workflows showed similar peptide and protein identifications in the pooled sample. The uHLC workflow was easier to set up and complete, having less hands-on time than the OM method, with fewer manual processing steps. Lower peptide and protein coefficient of variation was observed in the uHLC technical replicates. Following statistical analysis, candidate protein markers were filtered, at ≥8.35-fold change in abundance, ≥2 unique peptides and ≤1% false discovery rate, and revealed 121 significant, differentially abundant proteins, some of which have known associations with kidney injury. The pilot data derived using this novel workflow provide information on the urinary proteome of patients with AKI. Further exploration in a larger cohort using this novel high-throughput method is warranted.

## 1. Introduction

The study of the human urinary proteome is becoming increasingly popular in clinical proteomics studies. Large volumes of samples are readily available with minimal invasiveness, and, in addition, soluble proteins and peptides derived from various tissues and organs are also filtered in the urine, which can reflect more general health problems [1,2,3,4].

Plasma was long considered the best biofluid choice for biomarker discovery studies. However, the main drawback is the high complexity of the proteome due to a large protein dynamic range [5,6,7]. Therefore, protein biomarkers often expressed in minute amounts are difficult to detect and analyze reproducibly without the use of extensive depletion and fractionation strategies that reduce the complexity of the plasma proteome [8]. In contrast, urine has a smaller dynamic range and relatively lower complexity and is therefore more suitable for current analytical technologies [2,8]. The composition of the urinary proteome is minimally affected by homeostatic mechanisms during urine formation, so proteins that are filtered into the urine may serve as markers for nephropathy and systemic changes [9,10]. However, urinary proteomic analysis has unique challenges, particularly in extracting soluble urinary proteins present in dilute concentrations [7]. Recent research has seen an increase in the number of methods developed for robust urinary proteomics. The reported methods are based on precipitation [11,12,13], concentration [14,15], and on-membrane protein capture [16,17,18], thereby removing interfering compounds found in normal urine such as salts and other metabolites. The most common methods include acetone precipitation, trichloroacetic acid precipitation, ultracentrifugation, filter-aided sample preparation (FASP), and various combinations thereof [11,19,20,21]. After precipitation, protein resolubilization is often performed using urea-based buffers instead of more efficient detergent-based buffers, as detergent removal has historically been difficult to achieve [22]. A potentially high-throughput method, 96DRA-Urine, was recently developed and can accommodate 96 samples in parallel; however, it requires precipitation by acetone [23]. Berger et al. (2015) have developed a method, MStern Blot, that does not require precipitation and can process 96 samples in parallel. The method performed similarly to FASP in terms of protein and peptide coverage; however, it was significantly faster to complete. Unfortunately, there is no consensus on the ideal sample preparation methodology for urine processing and this remains the individual preference of the laboratory and is based on available resources. Furthermore, many of the current methods lack the throughput required to analyze large clinical cohorts due to bottlenecks created by steps such as precipitation, centrifugation, buffer exchange, and vacuum filtration, which are all difficult to scale and automate [13,23,24].

In the current study, we present a novel approach to the preparation of urinary proteome samples. The method, named urine-HILIC (uHLC), is based on direct, on-bead protein capture (from 100 µL of urine), clean-up, and digestion. It is automated (1 to 96 samples per run) and can be easily implemented in the mass spectrometry laboratory and requires standard sample collection procedures in clinics or hospitals. The HILIC-based workflow has been established for the clean-up and digestion of proteins from human tissue and plasma in previous studies [25,26], and here it has been modified for human urinary proteome sample preparation. The uHLC workflow was benchmarked against a urinary proteomics workflow based on on-membrane (OM) protein capture (MStern Blot) [16,17], as it is one of the more rapid and well-performing sample preparation methods amongst those established for urinary proteomics. A three by three approach was used to evaluate both workflows, that is, three technical replicates processed on three consecutive days (*n* = 9 per workflow). We then applied the uHLC workflow to an acute kidney injury (AKI) pilot study (*n* = 10) to show the applicability to typical proteomics research.

First-line antiretroviral therapy (ART) is freely available to 90% of people living with HIV in South Africa (~8.5 million total), accounting for ~20% of the global HIV/AIDS burden [27]. Most patients benefit from ART; however, it causes major side effects in others, and approximately 10% of African patients undergoing first-line ART experience AKI [28]. Current tests for AKI base diagnosis on elevated serum creatinine (sCr) levels [29,30], although this remains an unreliable marker of AKI in early or mild cases where sCr levels may remain normal despite kidney damage [31]. Therefore, more effective biomarkers are required to identify early renal dysfunction, allowing clinicians to make early interventions, such as a change in the ART regimen and closer monitoring of patients, to limit further ART-related nephrotoxicity. The analysis of urine for protein biomarkers related to kidney injury is ideal, as the composition of the urinary proteome can be influenced by glomerular filtration, tubular reabsorption, and tubular secretion [7,9,10], which can be directly affected by ART [32,33,34,35]. Using the uHLC workflow, we were able to show differentially abundant proteins and proteins known to be associated as disease markers for AKI. We show that the novel method reported is reproducible, robust, and efficient and has the potential to be used routinely in future clinical urinary proteomics research.

## 2. Materials and Methods

Solvents and chemicals (Mass spectrometry-grade) used in the study were purchased from MERCK unless otherwise specified. All buffers were freshly prepared. Sequencing grade modified trypsin was purchased from Promega (Madison, WI, USA). MagReSyn^®^ HILIC microspheres were purchased from ReSyn Biosciences (Edenvale, Gauteng, South Africa).

### 2.1. Urine Sample Collection Protocol and Pilot Study Cohort

Ethics approval was received for recruitment and collection of urine samples for this study (Ethics reference: #58/2013, #271/2018 (CSIR-REC) and #120612 (WITS-HREC)). For the development and benchmarking of the method, urine from three healthy adult men was used after informed consent (age range 26–38 years). Clinical samples were taken from unrelated patients who had been admitted to the Tshepong Hospital (Klerksdorp, South Africa). All participants were HIV-positive, African females, undergoing first-line combination ART (tenofovir-lamivudine-dolutegravir). They were age matched and grouped into AKI (case) and normal (control) based on their kidney function according to the guidelines set out in the Kidney Disease Improving Global Outcome report [36]. Briefly, AKI was confirmed clinically if patients showed one of the following: (a) increased serum creatinine ≥ 0.3 mg/dL within 48 h, (b) 1.5-times baseline that is known or presumed to have occurred in the last 7 days or (c) urine output < 0.5 mL/kg/h for 6 h. First-morning, midstream, clean-catch urine was collected into sterile urine collection containers and transported immediately on ice to prevent degradation. Individual samples were centrifuged at 800× *g* for 10 min to remove debris and then aliquoted and stored at −80 °C until further use.

### 2.2. Sample Preparation

#### 2.2.1. Automated Urine-HILIC Workflow

Samples were allowed to thaw to room temperature (RT). Urine (100 μL) was mixed with 300 uL of urine sample buffer (USB: 8M Urea, 2% SDS), and sequentially reduced and alkylated using dithiothreitol (DTT) (10 mM *v*/*v*; 30 min, RT) and iodoacetamide (IAA) (30 mM *v*/*v*; 30 min, RT-dark). Thereafter, an equal volume HILIC binding buffer (30% acetonitrile (MeCN)/200 mM ammonium acetate (NH_4_Ac) pH 4.5) was added to the sample-USB solution (~410 μL final volume) (Figure 1). The automated KingFisher™ HILIC workflow was then followed (protocol available from info@resynbio.com), with minor adjustments as described [25,26]. The automated on-bead protein capture, clean-up, and digest protocol was programmed using BindIt software v4.1 (Thermo Fisher Scientific, Waltham, MA, USA). Briefly, magnetic hydrophilic affinity microparticles (10 μL beads/100 μL urine) were equilibrated in 200 μL of 100 mM NH_4_Ac pH 4.5, 15% MeCN. The microparticles were then transferred to the well containing the sample-USB-bind buffer solution and mixed for 30 min at RT. The captured proteins were washed twice in 200 μL of 95% MeCN and transferred to 200 μL of 50 mM ammonium bicarbonate (ABC) containing 1 μg sequencing grade modified trypsin (Promega, Madison, WI, USA) and mixed for 2 h at 47 °C. Finally, beads were washed in 1% trifluoroacetic acid (TFA) to elute any remaining bound peptides. The resulting peptides (pool of digest and TFA eluate) were frozen at −80 °C and then dried at −4 °C using a CentriVap vacuum concentrator (Labconco, Kansas City, MO, USA), resuspended in 2% MeCN, 0.2% formic acid (FA) and quantified using the Pierce™ Quantitative Colourimetric Peptide Assay (Thermo Fisher Scientific, Waltham, MA, USA) according to the manufacturer’s instructions.

#### 2.2.2. On-Membrane Workflow Based on MStern Blot

The on-membrane protein capture protocol was used to benchmark the uHLC method, as it is an established method in urinary proteomics for large scale clinical research [16,17]. Briefly, 100 µL of urine was mixed with 300 µL of urea sample buffer (8 M urea in 50 mM ABC). Reduction with 30 µL reduction solution (150 mM DTT, 8 M Urea, 50 mM ABC) and alkylation with 30 µL (150 mM IAA, 8 M Urea, 50 mM ABC) were carried out at RT in the dark for 30 min each. Individual wells of polyvinylidene fluoride (PVDF) membrane plates (MSIPS4510, Merck Millipore, Burlington, MA, USA) were activated and equilibrated with 150 μL of 70% ethanol/water and urea sample buffer. Samples were passed through PVDF membranes using a vacuum manifold. Adsorbed proteins were washed twice with 150 μL of 50 mM ABC. Digestion was carried out at 37 °C for 2 h by adding 100 μL digestion buffer (5% *v*/*v* MeCN)/50 mM ABC) containing 1 μg sequencing grade modified trypsin per well. The plates were sealed with a sealing mat and placed in a humidified incubator, the resulting peptides were collected by applying vacuum and the remaining peptides were eluted twice with 75 μL of 40%/0.1%/59.9% (*v*/*v*) MeCN/FA/water. Samples were frozen at −80 °C and then dried at −4 °C using a CentriVap vacuum concentrator (Labconco, Kansas City, MO, USA). The samples were resuspended in 2% MeCN, 0.1% FA and then desalted using C18 StageTips according to the manufacturer’s instructions. Desalted peptides were frozen at −80 °C and then dried at −4 °C using a CentriVap vacuum concentrator. Finally, the peptides were resuspended in 2% MeCN, 0.2% FA and quantified using the Pierce™ Quantitative Colorimetric Peptide Assay (Thermo Fisher Scientific, Waltham, MA, USA) according to the manufacturer’s instructions.

### 2.3. LC SWATH-MS Data Acquisition

Individual peptide samples were analyzed using a Dionex UltiMate™ 3000 UHPLC in nanoflow configuration. Samples were inline desalted on an Acclaim PepMap C18 trap column (75 μm × 2 cm; 2 min at 5 μL/min using 2% MeCN/0.2% FA). Trapped peptides were gradient eluted and separated on a nanoEase M/Z Peptide CSH C18 Column (130 Å, 1.7 µm, 75 µm × 250 mm) (Waters Corp., Milford, MA, USA) at a flowrate of 300 nL/min with a gradient of 5–40%B over 30 min for benchmarking and 60 min for the pilot study (A: 0.1% FA; B: 80% MeCN/0.1% FA).

Data were acquired using data-independent acquisition (DIA)—or Sequential Window Acquisition of all Theoretical Mass Spectra (SWATH) [37], using a TripleTOF^®^ 5600 mass spectrometer (SCIEX, Framingham, MA, USA). Eluted peptides were delivered into the mass spectrometer via a Nanospray^®^ III ion source equipped with a 20 µm Sharp Singularity emitter (Fossil Ion Technology, Madrid, Spain). Source settings were set as: Curtain gas—25, Gas 1—40, Gas 2—0, temperature—0 (off) and ion spray voltage—3200 V.

Data were acquired using 48 MS/MS scans of overlapping sequential precursor isolation windows (variable *m*/*z* isolation width, 1 *m*/*z* overlap, high sensitivity mode), with a precursor MS scan for each cycle. The accumulation time was 50 ms for the MS1 scan (from 400 to 1100 *m*/*z*) and 20 ms for each product ion scan (200 to 1800 *m*/*z*) for a 1.06 sec cycle time.

### 2.4. Data Processing

A spectral library was built in Spectronaut™ 17 software using default settings with minor adjustments as follows: segmented regression was used to determine iRT in each run; iRTs were calculated as median for all runs; the digestion rule was set as “Trypsin” and modified peptides were allowed; fragment ions between 300 and 1800 *m*/*z* and peptides larger than 3 amino acids were considered; peptides with a minimum of 3 and maximum of 6 (most intense) fragment ions were accepted. This study-specific spectral library was concatenated with an in-house generated urinary proteome spectral library (using Spectronaut™ “Search Archives” feature).

Raw (.wiff) data files were analyzed using Spectronaut™ 17. The default settings that were used for targeted analysis are described in brief as follows: dynamic iRT retention time prediction was selected with correction factor for window 1; mass calibration was set to local; decoy method was set as scrambled; the false discovery rate (FDR), based on mProphet approach [38], was set at 1% on the precursor, peptide and protein group levels; protein inference was set to “default” which is based on the ID picker algorithm [39], and global cross-run normalization on median was selected. The final urinary proteome spectral library (peptides—20,616, protein groups—2604) was used as a reference for targeted data extraction.

Spectronaut™ 17’s default settings were used for state comparison analysis using a *t*-test (null hypothesis that no change in protein abundance was observed between the two groups). The *t*-test was performed on the log_2_ ratio of peptide intensities that corresponded to individual proteins. The *p*-values were corrected for multiple testing (Storey method) using the q-value approach to control FDR [40]. A retrospective power analysis was performed using the MSStats package (Northeastern University, MSStats 4.4.1) [41] in R (v 4.1.0) (Posit, Boston, MA, USA).

### 2.5. Bioinformatic and Clincial Data Analysis

Method development data, from each workflow, were acquired for three replicates on three consecutive days (*n* = 9). Peptide and protein coefficient of variation (CV) data were exported directly from Spectronaut™ 17 and plotted in GraphPad Prism (v9). Protein and peptide identification data were imported into ExPASy pI/MW [42] and GRAVY calculators. Protein data were further analyzed in Spectronaut™ 17 and exported into Microsoft Excel (v2305) to assess proteome coverage abundance scores (dynamic range assessment).

Patient clinical characteristic data were imported into GraphPad Prism (v9) and analyzed using Mann–Whitney tests with adjusted *p*-values as appropriate (*n* = 10). Where clinical data were missing, data were inferred using the mean value for the variable from the entire cohort. Protein abundance data were analyzed in ClustVis [43] and Enrichr [44] for principal component analysis (PCA) and gene ontology (GO) analysis, respectively. The volcano plot was plotted using http://www.bioinformatics.com.cn/srplot (accessed on 12 July 2023), an online platform for data analysis and visualization. All other graphs were generated in GraphPad Prism (v9).

## 3. Results

### 3.1. Workflow Time Comparisons

Both workflows required a similar total time to complete from start to finish (218 min uHLC vs. 205 min OM). However, the hands-on time was 15 min for the uHLC method and 60 min for the OM workflow (Figure 2G).

### 3.2. Peptide Yield

The workflows showed different peptide recoveries as shown in (Figure 2A). The OM workflow showed a mean peptide recovery of 0.16 µg peptide/µL urine (16.3 µg total). The uHLC workflow had a higher mean peptide recovery of 0.26 µg peptide/µL urine (26.0 µg total). For both workflows, a total of 500 ng of peptide was injected for LC-MS analysis based on colorimetric peptide assay calculations.

### 3.3. Peptides and Proteins Identified

The uHLC workflow had higher reproducibility than the OM workflow, as shown in the lower CVs at the protein level (Figure 2B), with median CV of 15.6–20% in the uHLC and 28–34.7% in the OM workflows, respectively. Similarly, at the peptide level (Figure 2C), median CV of 20.2–24.7% in the uHLC and 36.2–44% in the OM workflows were observed. PCA analysis also showed a tighter clustering of technical replicates in the uHLC workflow, indicating improved reproducibility compared to the OM workflow (Figure 2F). The workflows showed a similar total protein and peptide coverage across all replicates. A large overlap was observed between the two methods, with 7711 and 7477 peptides identified (Figure 2D), which corresponded to 1141 and 1070 protein identifications for the uHLC and OM workflows, respectively (Figure 2E) (Appendix A).

### 3.4. Protein Properties and Dynamic Range Comparison

The protein GRAVY score, molecular mass, and isoelectric point distributions were similar between both methods, showing little to no biases (Figure 3A–C). The protein isoelectric point showed a slight difference in the number of proteins recovered below a pI of 8, where uHLC showed a greater overall recovery. The uHLC workflow appeared to identify more proteins (16% vs. 12%) in the lower abundance range than the OM workflow (Figure 3D).

### 3.5. Pilot Study Clinical Data

No significant differences in age and phosphaturia were observed. Serum creatinine, estimated glomerular filtration rate (eGFR) and urine protein-to-creatinine ratio (UPCR) were significantly different between normal and AKI patients. sCr and UPCR was on average >10 times higher in the AKI group and eGFR was on average ~7 times lower in the AKI group (Table 1).

### 3.6. Pilot Study: Data-Independent Analysis for Clinical Samples

The uHLC workflow was applied to a pilot cohort of 10 HIV positive female patients to determine the urinary proteome level correlation between first-line ART and kidney dysfunction. Participants were matched by age and race and grouped into AKI (case, *n* = 5) and normal (control, *n* = 5) based on kidney function. A total of 4249 ± 639 and 5627 ± 1051 peptides were identified in the AKI and normal samples, respectively. These corresponded to 892 ± 92 and 1077 ± 138 proteins in the respective groups (Appendix A).

Following a *t*-test and a retrospective power analysis (α = 0.05, β = 0.8, Appendix A), only proteins with a fold change of ≥8.35 were considered significant (q value ≤0.01, ≥2 unique peptides). Using these inclusion criteria, 121 proteins showed differential abundance between normal and AKI patients (Figure 4A) (Supplementary File S2: candidate protein lists).

Data analysis of the urinary proteome revealed the presence of many proteins reported in the literature as candidate biomarkers of renal dysfunction. Selected known markers showed differential abundance between cases and controls (Figure 4A,B). The PCA analysis showed a distinct clustering of the limited number of AKI and normal participants based on quantitative proteomic data (Figure 4C).

## 4. Discussion

Urine has become an attractive biofluid source for biomarker studies because the proteome is less complex than biofluids that are used more commonly, such as plasma [1,2,8]. Successful biomarker studies require workflows that can be robust, easily implemented, and have high reproducibility.

A generally accepted approach to urinary protein sample preparation for mass spectrometry-based proteomics is precipitation-based. After the precipitation of urinary proteins, protein resolubilization can be difficult to achieve and often requires the use of strong detergents and/or salts that are not compatible with downstream mass spectrometry analysis [45]. Urinary proteomics studies commonly use organic solvent precipitation followed by FASP as a preferred method for the isolation, clean-up, and digestion of urinary proteins [46,47,48], and although this is a widely used and relatively simple procedure to follow, it is a laborious process and is prone to sample loss. This is due to numerous handling steps that also have the potential to introduce sample contamination. After FASP, samples need further processing, such as desalting and drying, before being analyzed, substantially increasing cost and time and perhaps more importantly a decrease in sample recovery. These shortcomings make urinary proteome analysis using organic solvent precipitation a complicated, cumbersome, and tedious process with low reproducibility. The latest developments in high-throughput LCMS workflows, using shorter LC gradients coupled to fast scanning mass spectrometers and DIA, allow for screening of a significantly higher number of samples. Thus, the emphasis has shifted to sample preparation to keep up with faster data acquisition. To this end, 96-well format methods have been developed, such as MStern, which can accommodate many samples in parallel and has been shown to perform better than FASP for urinary proteomics sample preparation [17]. This is a highly successful method; however, it lacks reproducibility, mainly due to its many manual steps, and the workflow cannot be easily automated, thus limiting its use.

In contrast, we present a novel sample processing method, urine-HILIC, that uses a small volume of urine (100 µL) mixed with urea and sodium dodecyl sulfate sample buffer with subsequent protein capture, clean-up, and on-bead digestion, using MagReSyn^®^ HILIC microspheres. The method shows performance similar to that of well-established methods, such as MStern, in terms of peptide and protein identifications. The physicochemical properties and dynamic range of the proteins identified using both methods were similar, although some method-specific biases were observed, as expected. The MStern workflow has already been shown to be approximately four times faster than more established methods such as FASP, mainly due to long centrifugations between steps that are essential in the FASP workflow [17]. In the current study, we showed that the uHLC method performed better than the MStern workflow in terms of speed and reproducibility. This is largely due to the minimal handling steps and the fact that uHLC is automated with significantly less hands-on time, especially for larger cohorts. This becomes increasingly important when large cohorts are analyzed where multiple rounds of pipetting and vacuum filtration, with varying rates of filtration per well, may lead to increased technical variability and potentially lower throughput. Furthermore, the uHLC method appeared to capture more proteins in the low abundance range, which may be highly relevant in biomarker discovery studies.

In the pilot cohort of HIV patients, we were able to detect differences in the urinary proteomes of the patients and many proteins that have been reported in the literature as markers for various forms of kidney damage. Selected differentially abundant proteins identified strongly correlate with those in the literature. Beta-2-microglobulin (B2MG_HUMAN) [49,50,51], and cystatin c (CYTC_HUMAN) [50,52,53] showed elevated urinary levels in patients with acute renal failure. A similar observation was made in kidney transplant patients who suffered rejection or postoperative renal complications in which pigment epithelium-derived factor (PEDF_HUMAN) increased in urine after surgery [34]. Similarly, patients in our cohort who suffered kidney damage expressed higher levels of these three proteins in their urine. Uromodulin (UROM_HUMAN), the protein most abundantly expressed in the urine of healthy patients [54,55,56], decreased significantly in our patients with kidney injury, possibly due to tubular damage leading to decreased excretion into the tubular lumen that contains urine [57]. This finding is important in kidney injury associated with first-line ART, as it is postulated that kidney injury is due to the accumulation of tenofovir in proximal tubule cells leading to toxicity [33,58,59,60]. Quintana et al. (2009) reported a similar result in which patients experiencing kidney damage expressed lower levels of uromodulin in their urine [61]. A strong enrichment of endopeptidase proteins was observed in patients with AKI, which is consistent with other studies in which these protein families showed associations with kidney injury [62].

The workflow comparison presented here is not an exhaustive assessment of all the methods currently used for urinary proteomics sample preparation; therefore, the conclusions are restricted. The limitations of the pilot study presented include a small sample size (which limits power) and confounders such as non-standard sample collection time and the presence of AKI in cases, which collectively limit the conclusions that can be drawn from the data. Preliminary data from this pilot study suggests that more exploration is needed, in a large and well-controlled cohort, to derive truly biologically meaningful findings.

## 5. Conclusions

We have developed a workflow, urine-HILIC, suitable for low-volume, direct, automated processing of clinical urine samples without the need for centrifugation or precipitation. The workflow shows promise for use in future urinary proteomics research and is simpler and faster, requiring less hands-on time than other workflows while maintaining the depth of coverage of the proteome. Furthermore, by applying the method in a pilot cohort, we were able to detect clinically relevant changes in the urinary proteome that are commonly associated with acute kidney damage. We have shown that the method is well suited for urinary proteome profiling and can be easily scaled for high-throughput clinical proteomics studies.

## Figures and Tables

**Figure 1 proteomes-11-00029-f001:**
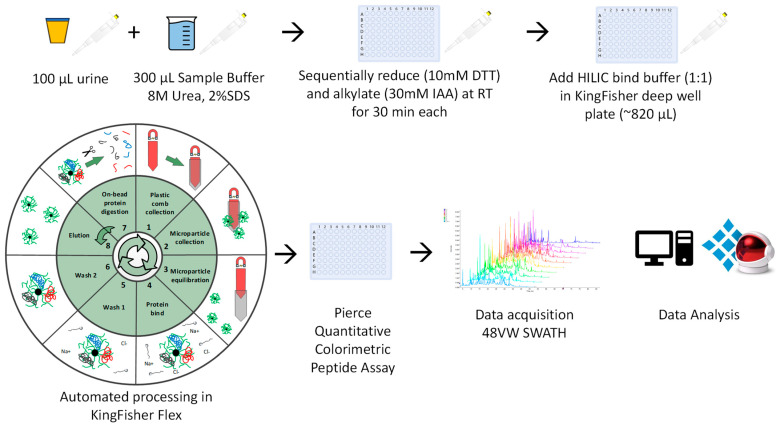
Schematic overview of the uHLC workflow.

**Figure 2 proteomes-11-00029-f002:**
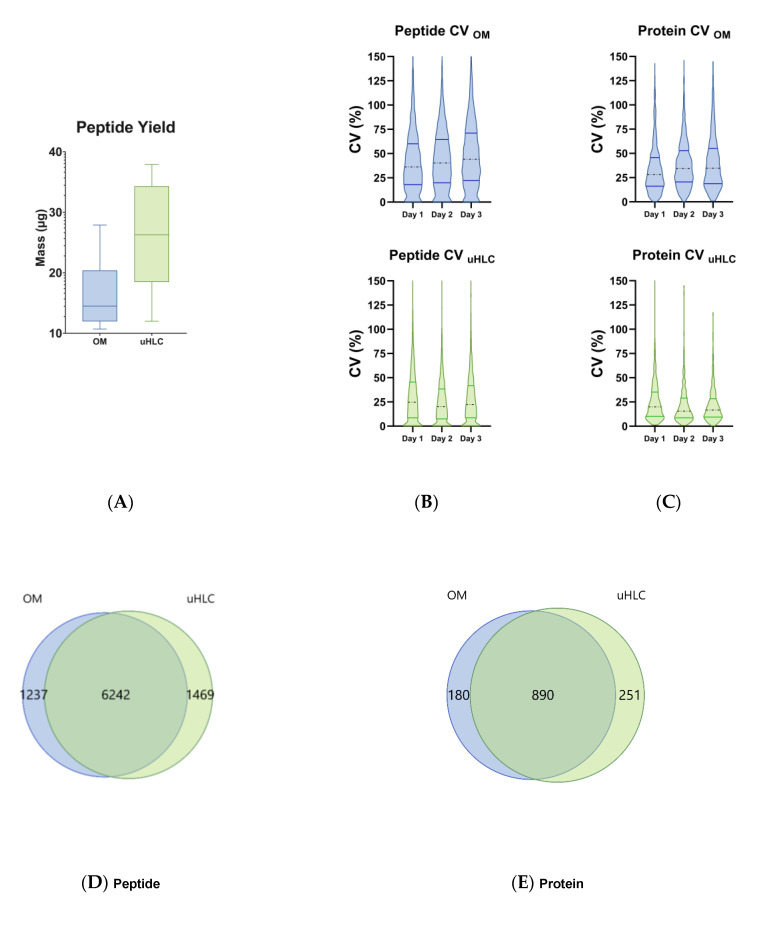
Yield and CV analysis between methods. (**A**) Total peptide recoveries of each method. uHLC shows lower CV at the peptide (**B**) and protein (**C**) levels for all technical replicates over three days. Venn diagram (**D**,**E**) showing similar protein and peptide identifications that were observed. (**F**) PCA plot shows tighter clustering of uHLC samples, indicating lower CV between technical replicates. (**G**) Time comparison showing a similar total time between workflows and a 4-times lower hands-on time for the uHLC method.

**Figure 3 proteomes-11-00029-f003:**
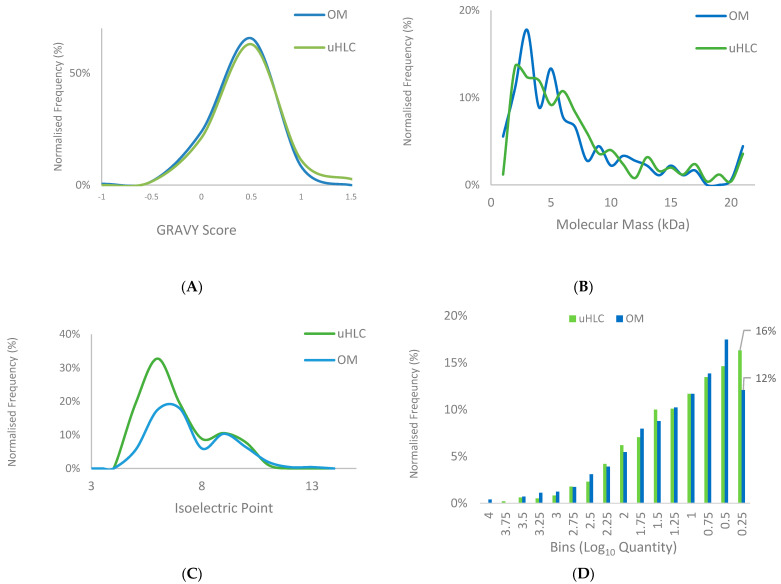
Protein properties and dynamic range comparison. (**A**–**C**) Protein level analysis of GRAVY score, molecular weight distribution and isoelectric point comparing uHLC (green) and OM (blue). (**D**) Protein abundance scores are displayed in discrete bins, from high (left) to low abundance (right).

**Figure 4 proteomes-11-00029-f004:**
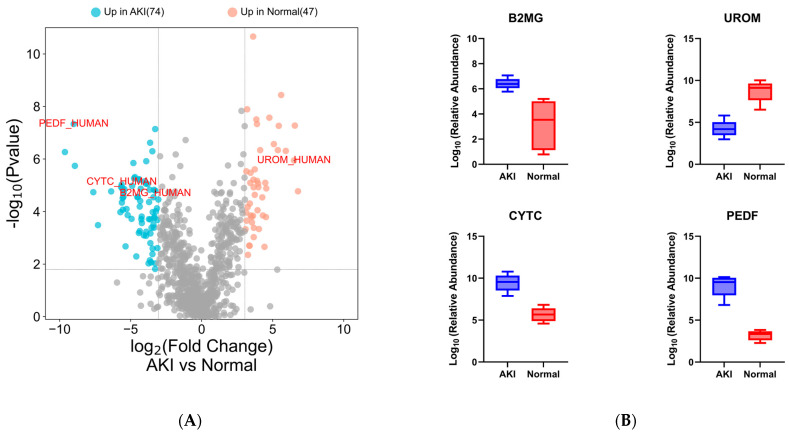
Differential analysis of pilot clinical proteomes. (**A**) Volcano plot showing differentially abundant proteins (≥8.35-fold change, ≤0.01% q-value, ≥2 unique peptides) including candidate protein markers (up in blue and down in red) and (**B**) known markers for kidney injury: PEDF, B2M, CYTC and UROM. (**C**) PCA plot; the X and Y axes show PC1 and PC2 that explain 22.8% and 20.1% of the total variance, respectively. (**D**) GO molecular function bar plot showing strong endopeptidase enrichment for the differentially abundant proteins (*p*-value ranked, adjusted *p* < 0.01).

**Table 1 proteomes-11-00029-t001:** Clinical characteristics for common parameters used to assess kidney function of participants included in the pilot study. All values reported as mean ± standard deviation. ns = not significant, *p* < 0.05 = significant.

Characteristic	Normal (*n* = 5)	AKI (*n* = 5)	*p* Value
**Age (years)**	35.4 ± 6.6	42.4 ± 12.5	ns
**Serum Creatinine_Admission_ (µmol/L)**	53.6 ± 4.17	563 ± 213.9	0.03
**Estimated glomerular filtration rate** **(mL/min/1.73 m^2^)**	108.6 ± 25.97	14.5 ± 11.8	0.03
**Urine Phosphate (mmol/L)**	1.62 ± 0.89	1.68 ± 0.93	ns
**Urine protein:creatinine ratio (g/mmol creat)**	0.025 ± 0.008	0.322 ± 0.18	0.03

## Data Availability

The mass spectrometry proteomics data have been deposited to the ProteomeXchange Consortium via the PRIDE [63] partner repository with the dataset identifier PXD043925.

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
