# Peer review of "Urine-HILIC: Automated Sample Preparation for Bottom-Up Urinary Proteome Profiling in Clinical Proteomics"

_proteomes, 2023, doi:10.3390/proteomes11040029_

Round 1

Reviewer 1 Report

Govender and colleagues submitted a manuscript entitled “Urine-HILIC: Automated sample preparation for bottom-up urinary proteome profiling in clinical proteomics” for publication as Technical Note in Proteomes. The manuscript may certainly be of interest, but some issues need attention, as further outlined below. This is even more important in this specific case as the authors are employed be the vendor of the presented technological approach, hence there is a clear conflict of interest (which should not prevent publication of the results).

1)      The experimental data are not shown, just results or conclusion thereof (e.g. CV, % of low abundance proteins, etc.). I suggest submitting a supplementary table that lists the peptides identified and the proteins, as well as the relative abundance values (counts). In this way, the reader can easily follow the results presented. In addition, such a table would be helpful when comparing to other reports.

2)      Methods are not always clear: the authors report recovery of 16 or 26 µg peptide per 100 µl urine, but then they inject 500 µg peptide. Were multiple (20-35) preparations combined for one LC-MS/MS run?

3)      The CVs and other data presented are based on how many replicates?

4)      The pilot study on AKI is missing quite some information. Please provide a table with the relevant clinical and demographic information, including confounders (e.g. proteinuria, hematuria, eGFR, etc.).  How was the diagnosis of AKI established? What was the cause of AKI (nephrotoxicity of the drugs?)? When was the urine sampled (time of day and also with reference to AKI diagnosis, where applicable). I fear that this study is substantially underpowered, upon adjustment for the major confounders, which is absolutely mandatory, likely no significant findings can be observed (I expect that many of the findings in the unadjusted data are the result of proteinuria in the AKI patients). This should not preclude showing the data, but obviously would not allow for any claims of potential biomarkers.

5)      With reference to the issue mentioned above, this obviously has a major impact on Figure 4

6)      The authors should present the current state of the art in some more detail. E.g. there are also other generally accepted protocols for urine sample preparation like SPE or desalting. Along these lines, the authors state that their protocol is simpler and faster than the other workflows, but the evidence for this statement is not give. It seems very helpful to show a figure that graphically depicts the different workflows in a comparative way, including the time requirements.

A minor issue: I think its molecular mass, not molecular weight

Based on the above, the paper apparently needs quite some work before it could be considered for publication in a leading journal in the field. At the same time, a substantially revised version would likely be of interest to the readers.

Reviewer 2 Report

Overall a good study for future urine proteome studies. But these is something important feature that the authors missed. Urine is not controlled by homeostatic mechanisms of the body and therefore reflects earliest and most subtle changes. This is extremely important for the medicine in the future. I hope that authors can add it in the background.

Reviewer 3 Report

In this manuscript, the authors describe an automated method for processing urine samples based on the use of MagReSyn® HILIC microspheres. The topic is quite interesting and relevant since the proteomics field is rapidly moving to a new phase, in which the projects require the analysis of large clinical cohorts, which is impossible to manage manually. The manuscript is well-written and has a lot of potential but clearly, there are a lot of aspects that must be improved, mainly in the section of the results.

Introduction

·  Line 44-46, pages 1-2: “However, urinary proteomic analysis has unique challenges, particularly in extracting soluble urinary proteins present in dilute concentrations.”

Please add some references to this sentence. For example:

· Kalantari, Shiva, Ameneh Jafari, Raheleh Moradpoor, Elmira Ghasemi, and Ensieh Khalkhal. 2015. “Human Urine Proteomics: Analytical Techniques and Clinical Applications in Renal Diseases.” International Journal of Proteomics 2015. https://doi.org/10.1155/2015/782798.

· Gao, Youhe. 2015. “Urine Is a Better Biomarker Source Than Blood Especially for Kidney Diseases.” In Urine Biomarker Kidney Disease Proteomics in Discovery, 3–12. https://doi.org/10.1007/978-94-017-9523-4_1.

·  Zou, Lili, and Wei Sun. 2015. “Human Urine Proteome: A Powerful Source for Clinical Research.” In Urine Biomarker Kidney Disease Proteomics in Discovery, 31–42. https://doi.org/10.1007/978-94-017-9523-4_4.

· Line 54-59, page 2: “Unfortunately, there is no consensus on the ideal sample preparation methodology (…) created by steps such as precipitation, centrifugation, and buffer exchange, which are all difficult to scale and automate..”

Please add some references to this paragraph.

· Line 54-59, page 2:We then applied the novel workflow to an acute kidney injury (n = 12) pilot study to show applicability to typical proteomics research”.

Although I understand the main goal of this work is the description of the developed method, I think that will be important to include 1 or 2 sentences about AKT and its relation with first-line ARV therapy. As a matter of fact, I only understood the purpose of the pilot study in the results section. Also, it will be important to explain why urine is a relevant source of specific biomarkers of kidney damage. Something like: urine composition is highly affected by problems at the level of glomerular filtration rate, tubular reabsorption, etc., as the majority of urinary proteins are secreted from tubules and kidney-specific cells. The authors can use the above-mentioned bibliography.

Materials and Methods

·      Line 91-92, page 2: “No protease inhibitors were used in this study.”

Usually, the authors refer to protease inhibitors if it was added to the samples but it is curious that here the authors refer specifically that “No protease inhibitors were used in this study.” Why?

· Line 100-102, page 3: “Urine (100 μL) was mixed 96 with 300 uL of urine sample buffer (USB: 8M Urea, 2% SDS), and sequentially reduced 97 and alkylated using DTT (10mM v/v; 30 min, RT) and IAA (30mM v/v; 30 min, RT-dark).”

I was curious about the motive of the use of urea in the sample buffer. Since the proteins in urine are already in solution, I think it would not be required such as a harsh buffer (including the fact that in the majority of cases, SDS is enough to make a good lysis/extraction).

· Line 100-102, page 3: “ The automated KingFisher™ HILIC workflow was then followed (protocol available from info@resyn-101 bio.com), as previously described [14,15].

The authors claim in the introduction that “In the current study, we present a novel approach to the preparation of urinary proteome samples.” (lines 60-61, page 2). However, in the materials and methods, they state that the automated protocol was then followed as previously described. So, the method was already been described and applied for the preparation of other sample types in other publications (that I assume are from the same research group) and for the preparation of urine samples ( protocol from ReSyn Biosciences with ID: HILIC-RAPOBD_URINE_2). I compared the methods used in the current work with the ones applied in these publications and ReSyn Biosciences protocol and I understand that minor adjustments were made (for example, protein:bead ratio). Also, compared to the method described in the ReSyn Biosciences protocol,  the workflow described here allows to directly process urine samples (which is referred to briefly in the introduction in lines 61-62), without the requirement of acetone precipitation, This is relevant since this workflow reduces processing time and enables automated sample processing for large sample cohorts.

So, the authors should clarify this fact in the introduction and refer to that adjustments were made compared to already published methods. Also, the authors should refer to “that KingFisher™ HILIC workflow was then followed (protocol available from info@resyn-101 bio.com), with minor adjustments. 

·  Line 64-66, page 2: “The uHLC workflow was benchmarked against a urinary proteomics workflow based on on-membrane (OM) protein capture (MStern approach) [9,10], a high throughput method that can accommodate 96 samples in parallel.” and Lines 118-119, page 3: “ The on-membrane protein capture protocol was used to benchmark the uHLC method, since it is an emerging method in urinary proteomics for large scale clinical research [9,16].”

In addition to the MStern approach, the authors did some comparison with the MAG HILIC urinary proteomics workflow (HILIC-RAPOBD_URINE_2) or are thinking about it? I recently read the HUPO poster “Rapid and automatable magnetic microparticle-based methods for clinical proteogenomic studies” in which several proteomics workflows based on magnetic-beads were compared, but of course, the results of the current method were not included.

·  Line 162-163, page 4: “ions with greater than 3 amino acids were considered”

Usually, peptides larger than 3 amino acids were considered for the analysis.  

·  Line 176-179, page 4: “Default settings were used for state comparison analysis using a t-test (null hypothesis that no change in protein abundance was observed between the two groups). The t-test was performed on the log2 ratio of peptide intensities that corresponded to individual proteins. The p-values were corrected for multiple testing using the q-value approach to control false discovery rate [20].”

The authors refer to the statistical test but do not refer to how the statistical analysis was performed (e.g. software). Also, they did refer to the method used for correcting the multiple testing (e.g. Benjamini–Hochberg method).

·      Section “2.4. Data processing”, page 4:

It is recommended to upload the proteomics data to repositories such as ProteomeXchange (https://www.proteomexchange.org/).

Results

·  Line 191-192, page 4: “The workflows showed different peptide recoveries as shown in (Figure 2A). OM workflow showed a mean peptide recovery of 0.16 μg peptide/μL urine (16.3 μg total). The uHLC workflow had a higher mean peptide recovery of 0.26 μg peptide/μL urine (26.0 μg total).”

Did the authors calculate the yield? From the description of the method, I think that the urine samples were not quantified before processing.

·  Line 194-195, page 4: “For both workflows, a total of 500 μg of peptide was injected for LC-MS 194 analysis based on colorimetric peptide assay calculations.”

Please, correct the peptide quantity to 500 ng.

·  Line 197-200, page 5: “The uHLC workflow had higher reproducibility than the OM workflow, as shown in the lower CVs at the protein level (Figure 2B), with median CVs of 15.6% – 20% in the uHLC and 28% – 34.7% OM workflows, respectively. Similarly, at the peptide level (Figure 2C), median CVs of 20.2% – 24.7% in the uHLC and 36.2% – 44% OM workflows, were observed.” and Figure 2B/C (Page 6).

Regarding the peptide identification, even though the optimized method has a lower CV than OM workflow, it will be recommended to have a CV lower than 20 %, while a CV ranging from 20% to 30% is considered acceptable. Another point that is a little bit confusing is the graphic representation of the %CV, which makes me wonder how the CV value was calculated. The CV is usually calculated by the dividing standard deviation by the mean. In my opinion, it would be more adequate to represent in the graph the number of peptides and proteins identified in the sample. As the violin plots are a good representation of the distribution of numeric data, it will give an idea of the variability of the data.

·  Figure 3, page 7:

Correct “normalised frequency (%)” in Figure 3D.

·  Section “Pilot study: data-independent analysis for clinical samples”, page 7:

The results of the proteome analysis of the urine from 12 HIV patients were not properly described. There is a lot of information missing in the manuscript, namely the number of proteins (protein groups) identified, quantified, and found differentially expressed between groups, although some of this information is described the Figure 4A. Also, the results of the quantitive proteomics analysis are usually included as supplementary material.

Also, demographic characteristics of patients involved in the study were not included in the manuscript, it was only described that participants were matched by age, race, and gender in both groups. The authors also intend to determine if there was a correlation between first-line ARV treatment and kidney dysfunction. However, the authors only referred to that kidney function was assessed according to the guidelines in the Kidney Disease Improving Global Outcome report (lines 88-89, page 2), but no clinical values were shown.

Discussion

The first part of the discussion is adequate, although the peptide yield and CV obtained with this method, as well as the proteins identified, could have been compared to the ones obtained with other methods (studies in the literature).

The paragraph on potential biomarkers of kidney damage may be expanded upon but I recognize that this was not the main objective of this work.

Round 2

Reviewer 1 Report

Govender and colleagues have submitted a revised manuscript entitled “Urine-HILIC: Automated sample preparation for bottom-up urinary proteome profiling in clinical proteomics”.

The paper is substantially improved and the authors have addressed all previously raised comments.

Some additional changes seem to be required to avoid misleading readers. This is even more important given the conflict of interest of several authors:

A section on the limitation of the study is missing, which should also help avoiding giving the impression that the platform presented has solved the issues the authors indicate: lack of reproducibility of other platforms. First, several sample preparation methods have demonstrated very good reproducibility and extensive data have been published on the performance. At least some of these references should be listed to present the current state-of-the-art in an objective way. Second, the assessment of reproducibility presented here is at best preliminary, but insufficient for a solid assessment (please have a look at the relevant guidelines from regulators or respective professional societies). Also, quite frankly, the CVs are very high, but this may be improved upon a more thorough investigation of the performance.

I also strongly suggest indicating in the limitation section that the pilot study presented in the 10 subjects was in no way intended to identify any potential biomarkers for AKI, only to demonstrate that potential differences in samples can be detected. The lack of power, presence of confounders and the study design (samples were collected at unspecified timepoint from subjects with established AKI, collection of first morning urine) all make this dataset inappropriate for the identification of potential biomarkers. Along these lines, many of the expected biomarkers apparently were not detected, at least not as changed in AKI (e.g. NGAL, KIM1, etc.).

Reviewer 3 Report

The authors significantly improved the manuscript. I am pleased with the changes. 

Author Response

Thank you, again, for the positive feedback and for helping us improve the final version of the paper.